# Lung Ultrasound Improves Outcome Prediction over Clinical Judgment in COVID-19 Patients Evaluated in the Emergency Department

**DOI:** 10.3390/jcm11113032

**Published:** 2022-05-27

**Authors:** Paolo Bima, Emanuele Pivetta, Denise Baricocchi, Jacopo Davide Giamello, Francesca Risi, Matteo Vesan, Michela Chiarlo, Giuliano De Stefano, Enrico Ferreri, Giuseppe Lauria, Stefano Podio, Peiman Nazerian, Franco Aprà, Enrico Lupia, Fulvio Morello

**Affiliations:** 1Medicina d’Urgenza—MECAU, Presidio Molinette, A.O.U. Città della Salute e della Scienza di Torino, 10126 Torino, Italy; paolo.bima@edu.unito.it (P.B.); emanuele.pivetta@unito.it (E.P.); francescarisi41290@gmail.com (F.R.); met88.vesan@gmail.com (M.V.); enrico.lupia@unito.it (E.L.); 2Scuola di Specializzazione in Medicina d’Emergenza-Urgenza, Università degli Studi di Torino, 10126 Torino, Italy; denisebaricocchi@gmail.com (D.B.); jacopo.giamello@gmail.com (J.D.G.); spodio@ausl.vda.it (S.P.); 3Dipartimento di Emergenza e Accettazione, A.O. Parini, 11100 Aosta, Italy; 4Medicina d’Urgenza, A.O.S. Croce e Carle, 12100 Cuneo, Italy; lauria.g@ospedale.cuneo.it; 5Medicina e Chirurgia d’Accettazione e d’Urgenza, Ospedale San Giovanni Bosco, 10154 Torino, Italy; michelachiarlo@gmail.com (M.C.); franco.apra@aslcittaditorino.it (F.A.); 6Medicina e Chirurgia di Urgenza e Accettazione, A.O.U. Careggi, 50134 Firenze, Italy; giul.destefano@gmail.com (G.D.S.); pnazerian@hotmail.com (P.N.); 7Medicina e Chirurgia d’Accettazione e d’Urgenza, Ospedale Maria Vittoria, 10144 Torino, Italy; enrico.ferreri@aslcittaditorino.it; 8Dipartimento di Scienze Mediche, Università degli Studi di Torino, 10126 Torino, Italy

**Keywords:** COVID-19, prognosis, score, mortality, disposition, lung ultrasound

## Abstract

In the Emergency Department (ED), the decision to hospitalize or discharge COVID-19 patients is challenging. We assessed the utility of lung ultrasound (LUS), alone or in association with a clinical rule/score. This was a multicenter observational prospective study involving six EDs (NCT046291831). From October 2020 to January 2021, COVID-19 outpatients discharged from the ED based on clinical judgment were subjected to LUS and followed-up at 30 days. The primary clinical outcome was a composite of hospitalization or death. Within 393 COVID-19 patients, 35 (8.9%) reached the primary outcome. For outcome prognostication, LUS had a C-index of 0.76 (95%CI 0.68–0.84) and showed good performance and calibration. LUS-based classification provided significant differences in Kaplan–Meier curves, with a positive LUS leading to a hazard ratio of 4.33 (95%CI 1.95–9.61) for the primary outcome. The sensitivity and specificity of LUS for primary outcome occurrence were 74.3% (95%CI 59.8–88.8) and 74% (95%CI 69.5–78.6), respectively. The integration of LUS with a clinical score further increased sensitivity. In patients with a negative LUS, the primary outcome occurred in nine (3.3%) patients (*p* < 0.001 vs. unselected). The efficiency for rule-out was 69.7%. In unvaccinated ED patients with COVID-19, LUS improves prognostic stratification over clinical judgment alone and may support standardized disposition decisions.

## 1. Background

Pandemic waves of SARS-CoV-2 infection lead to Emergency Department (ED) overcrowding, hospital admission peaks for COVID-19 and shortages in hospital beds for non-COVID-19 patients [1,2,3]. The standardized identification of SARS-CoV-2 positive patients not requiring hospital admission is therefore a clinical and system need. Ideal candidates for ED discharge and home treatment are individuals with mild disease and a sufficiently low risk of adverse events such as respiratory failure and death. However, the development of standardized ED disposition rules for COVID-19 is challenging, because balancing the assessment of current clinical severity and the risk of subsequent deterioration is demanding [4].

Prognostication tools or scores evaluating key clinical variables may assist ED disposition decision, as already established for community-acquired pneumonia [5,6]. For instance, the HOME-CoV (HCR), developed and validated in the ED, dichotomically identifies patients at a low risk of adverse events based on clinical presentation, course, comorbidities and living context. The 4C mortality score (4CMS), instead, developed on large inpatient cohorts, provides a graded risk-stratification (score 0 to 21) based on clinical presentation, comorbidities and selected blood test results [7,8,9,10,11]. However, the superiority of standardized rules/scores over subjective clinical judgment has not been shown, and the potential effects of these tools on hospital admission rates are largely unknown.

In ED facilities, lung ultrasonography (LUS) has emerged as a key point-of-care or portable tool for assessment of COVID-19 patients, either suspected or confirmed [12,13,14]. At the patient’s bedside, LUS can easily and rapidly detect SARS-CoV-2-related interstitial lung involvement, and in COVID-19 patients, LUS findings have prognostic value, with extensive/severe patterns predicting an increased risk of ARDS, mechanical ventilation and death [15,16,17]. LUS could therefore provide valuable data to complement clinical judgment and to inform disposition decisions. The present study tested the hypothesis that LUS, alone or integrated with standardized clinical tools, may improve the risk stratification and disposition decision for COVID-19 patients evaluated in the ED, compared to clinical judgment alone.

## 2. Methods

### 2.1. Design and Setting of the Study

This was a multicenter prospective observational cohort study performed on COVID-19 patients from six EDs, in three Italian regions. This was a multicenter prospective observational cohort study performed on COVID-19 patients from six EDs, in three Italian regions. Two are located in large tertiary university hospitals (Molinette Hospital, Turin and Careggi Hospital, Florence), two in tertiary non-university hospitals (Santa Croce e Carle Hospital, Cuneo; San Giovanni Bosco Hospital, Turin) and two in secondary non-university hospitals (Maria Vittoria Hospitals, Turin; Parini Hospital, Aosta). Enrolment was conducted from 10 October 2020 to 11 January 2021. The study was registered on clinicaltrials.org (NCT04629183) and approved by the local ethics committee (Comitato Etico Interaziendale A.O.U. Città della Salute e della Scienza di Torino, A.O. Ordine Mauriziano, A.S.L. Città di Torino, 0031189/24 March 2020 and 0009810/29 January 2021). The general patient characteristics and validation of the 4CMS in this cohort have been previously described [11].

### 2.2. Characteristics of Participants

Consecutive ED outpatients were enrolled in the presence of (1) a positive molecular test for SARS-CoV-2 obtained within 14 days prior to the index visit; (2) COVID-19 symptoms leading to an ED visit and dating <14 days; (3) discharge disposition from the ED, owing to physician’s decision or patient’s own will. Exclusion criteria were age <18 years, refusal to give consent, nursing care residence, long-term oxygen therapy and previous ED access for suspected or confirmed COVID-19. Only patients subjected to LUS were analyzed in the present study.

### 2.3. Interventions during the Index Visit

The workup of eligible patients was independent of study participation. ED physicians operated in compliance with national guidelines from the Ministry of Health (MOH, Circ. 0024970-30 November 2020), but ED disposition decisions were based on subjective clinical judgment and did not follow a standardized protocol. All diagnostic results were available to treating physicians during the index visit. Attending physicians prospectively recorded demographic, clinical and LUS data on a standardized case report form.

### 2.4. Lung Ultrasonography

Attending physicians were all trained in LUS of COVID-19 patients [18]. If logistically feasible, they performed LUS as a point-of-care exam during the index visit. In order to expedite evaluation and reporting, they were instructed to perform a minimum standardized assessment using either a curvilinear transducer (5–3 MHz, Esaote Mylab5 or Mylab7, Genova, Italy) or a handheld device (Butterfly iQ; Butterfly Network Inc., Guiford, CT, USA) with a lung preset (3 MHz). Representative scans were registered for random quality control, but systematic recording and independent adjudication were not performed.

The thorax was scanned thoroughly for presence of interstitial syndrome and lung consolidations, as previously described [12]. A modified semi-quantitative LUS score was calculated as the score for B-lines (indicating interstitial lung inflammation) + score for lung consolidations. The score for B-lines was calculated as the number of lung areas with ≥3 B-lines, using 8 areas (4 per side, shown in Figure 1A,B). The score for lung consolidations was 0 for absence of consolidations, 3 for unilateral consolidation(s) or 6 for bilateral consolidations [19]. A lung consolidation was defined as evidence of a non-aerated lung tissue consolidation larger than 1 cm (Figure 1C) [20]. Hence, the modified LUS score ranged from 0 (absence of B-lines and consolidations) to 14 points (presence of B-lines in 8 areas plus bilateral consolidations).

### 2.5. Clinical Score Calculation

The HCR, described in Appendix A, was calculated according to Duillet et al., excluding the variable “clinically significant worsening within the last 24 h”, which lacked a standard definition [7]. The 4CMS, described in Appendix A, was calculated according to Knight et al. [8]. If urea was not available, we used corresponding creatinine cutoff levels established on a training ED cohort of 832 patients subjected to a simultaneous urea/creatinine assay, as previously detailed [11].

### 2.6. Outcomes

The primary outcome was as a composite of all-cause death or hospital admission occurring within 30 days from the index ED visit. The secondary outcome was a composite of all-cause death or the need for non-invasive ventilation, high-flow nasal cannula or intensive care unit admission, occurring within 30 days.

For outcome retrieval, we performed a hospital database search, acquisition of medical charts and a structured phone interview conducted by a trained researcher. Two independent and expert physicians, blinded to the index ED visit data, made final case adjudication. In case of discordance, a third independent evaluation was planned.

### 2.7. Study Power

The present study was powered to test the null hypothesis that a binary discharge rule based on LUS identifies patient groups with a 10% difference in primary outcome occurrence (5% in the LUS negative group and 15% in the LUS positive group). This estimate was based on previous data, where mortality in COVID-19 patients potentially suitable for ED discharge was 1.2% for low-risk patients and 9.9% for intermediate risk patients, according to 4CMS [8]. Using an alpha error of 5% and a power of 90%, we estimated that at least 300 patients needed to be included.

### 2.8. Data Analysis

We expressed continuous variables as the mean and standard deviation (SD) or median and interquartile range (IQR), as appropriate. Categorical variables were expressed as absolute numbers and percentages. We assessed prognostic discrimination for each outcome using the C-indexes, which were compared using the DeLong’s test for paired curves, and standard performance measures (sensitivity, specificity) [21]. The best cut-off for the LUS score was the one associated with the maximum product of sensitivity and specificity [22]. Sensitivity and specificity values of different strategies were compared using the binomial exact test (paired samples) [23]. Proportions of patients ruled-out with different strategies were compared using a z test for partially overlapping samples [24]. Overall goodness of fit for LUS results was assessed using the Brier score [25] and model calibration with Cox’s intercept and slope [26].

The survival analysis was carried out with the Kaplan–Meier estimator, using the log-rank test for comparison of the curves and Cox regression. *p*-values were considered statistically significant if <0.05. The statistical analysis was performed with R (v3.6.4).

## 3. Results

### 3.1. Patient Characteristics and Outcomes

Within 521 patients with enrolment criteria, 393 with available LUS data were further analyzed. The mean age was 51 ± 16 years. Two-hundred patients (50.9%) were male, and 79 (20.1%) had at least one comorbidity. One-hundred twenty patients (30.5%) were HCR positive, and 30 (7.6%) were at high risk, according to 4CMS. The overlap of HCR and 4CMS classification is shown in Appendix A.

The primary outcome occurred in 19 HCR-positive and in 8 high-risk patients (Appendix A). Within 30 days, the primary outcome was reached in 35 (8.9%) patients and the secondary outcome in 14 (3.6%). Fourteen (3.6%) needed NIV/HFNC, one (0.3%) was admitted to intensive care and two (0.5%) patients died.

### 3.2. LUS Results

LUS-defined lung involvement was absent (score = 0) in 234 (59.5%) patients, interstitial involvement in one lung area (score = 1) in 40 (10.2%) and in multiple lung areas or any consolidation (score ≥ 2) were present in 119 (30.3%, of whom 10 (8.4%) had ≥1 lung consolidation). As shown in Figure 2 and in Appendix A, increased LUS-defined lung involvement was associated with worse clinical outcomes. The calibration plots for LUS-based outcome prognostication are shown in Appendix A. The Brier score, Cox’s intercept and slope for the primary and secondary outcome were respectively 0.08/0/1, and 0.03/0/1.04, indicating good overall performance and modest calibration. The cross-tabulation of LUS and clinical scores is represented in Appendix A.

### 3.3. Outcome Prediction

The prognostic curves of clinical scores and LUS for outcome prediction are shown in Figure 3. For the primary outcome, C-index values were 0.63 (0.54–0.72) for HCR, 0.79 (0.73–0.86) for 4CMS (*p* < 0.001 vs. HCR) and 0.76 (95%CI 0.68–0.84, *p* = 0.04 vs. HCR) for LUS. For the secondary outcome, C-index values were 0.64 (95%CI 0.50–0.77) for HCR, 0.75 (95%CI 0.66–0.85) for 4CMS (*p* = 0.09 vs. HCR), and 0.8 (95% 0.68–0.92) for LUS (*p* = 0.11 vs. HCR). Logistic models including each diagnostic tool along with symptom onset showed similar results (Appendix A).

The sensitivity and specificity values of the LUS for outcome prediction are shown in Figure 3. The optimal LUS cut-off was 2 for both outcomes. A positive LUS (score ≥ 2) identified a subgroup of patients at increased outcome probability at the Kaplan–Meier estimator (log-rank test *p*-value < 0.001 for both outcomes, Figure 4). The hazard ratio of positive LUS adjusted for age and sex was 4.33 (95%CI 1.95–9.61) for the primary outcome and 5.75 (95%CI 1.48–22.3) for the secondary outcome.

### 3.4. Comparison of LUS with Clinical Scores

We next evaluated risk stratification obtained with LUS, HCR or 4CMS alone or the integration of HCR/4CMS with LUS. Sensitivity and specificity values for outcome prediction are shown in Table 1. For both outcomes, sensitivity was highest using HCR-LUS integration (HCR positive or LUS positive).

Outcome occurrence rates in patients satisfying different rule-out criteria are shown in Table 2. In LUS-negative patients, the primary and secondary outcome occurrence rates were 3.3% and 1.1%, respectively (*p* < 0.001 vs. unselected). The corresponding rule-out efficiency was 69.7%. The lowest outcome occurrence rates (1.4% primary, 0.5% secondary) were observed in HCR-negative and LUS-negative patients (*p* < 0.001 vs. LUS negative patients), with an efficiency of 52.7%. Rule-out using an integration of 4CMS with LUS was associated with occurrence rates of 2.7% and 1.2% and an efficiency of 65.9%.

## 4. Discussion

To our knowledge, this is the first study directly evaluating LUS as a tool assisting physicians for risk assessment of COVID-19 outpatients. Results show that LUS provides an incremental prognostication capacity over clinical judgment alone and that LUS-based discharge rules may improve the safety of ED discharge dispositions.

In this study cohort recruited in the pre-vaccination era, a non-standardized discharge disposition made by treating physicians through subjective clinical judgment/gestalt was indeed associated with a substantial number of patients (1 in 11) requiring subsequent hospital admission or developing adverse clinical outcomes, including death. This finding is in line with a previous North American study reporting 8.2% readmission rate at 7 days for unvaccinated COVID-19 patients and confirms the need for improved risk-stratification of COVID-19 patients in the ED when evaluating final disposition [27].

In the present study, negative LUS, defined by interstitial involvement in no more than one lung area and the absence of lung consolidations, was indeed associated with mild clinical outcomes (3.3% and 1.1% for primary and secondary, respectively). The expected impact of this approach on hospital admissions, compared to clinical judgment alone, is moderate, with about 30% of patients requiring hospital admission. A further reduction in adverse clinical outcomes can be obtained by integrating LUS with a clinical score (either HCR or 4CMS), leading to a more substantial increase in patients requiring hospital admission in association with HCR (about 50%).

Our study has limitations. First, we conducted the study in a pre-vaccination phase. Therefore, outcome estimates would now essentially apply to unvaccinated individuals. Second, the study’s focus on discharged patients likely led to the selection of milder and less frail patients, limiting external validity for unselected ED patients. Third, LUS was not performed in all patients, but the reason for not performing LUS was not reported by the attending physicians. Fourth, although LUS has a steep learning curve and LUS was always performed by trained physicians, LUS images were not recorded and did not undergo central adjudication [28]. Fifth, conduction of the study during a pandemic peak pragmatically led to calculation of a quick semi-quantitative LUS score. The corresponding cutoff using a standard LUS score is likely about 4, but external validation is warranted [29]. Finally, our study protocol did not include blinding, since LUS was performed during the medical evaluation by attending physicians, potentially influencing clinical management and final disposition.

## 5. Conclusions

In unvaccinated COVID-19 patients evaluated in the ED for final disposition, LUS assessment, alone or integrated with a clinical risk score, improves outcome prognostication over clinical judgment alone. Patients with minor or absent LUS findings are at a very low risk of adverse outcomes and can be safely discharged. The systematic application of LUS for the disposition decision is expected to increase the safety of patient discharge over clinical judgment alone, while slightly increasing hospital admission rates. Confirmatory studies on contemporary cohorts comprising vaccinated patients and new SARS-CoV-2 variants are warranted.

## Figures and Tables

**Figure 1 jcm-11-03032-f001:**
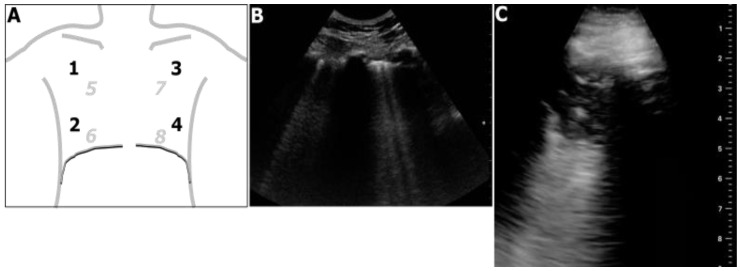
Panel (**A**): Lung areas for the calculation of the modified LUS score. 1: right upper antero-lateral area; 2: right lower antero-lateral area; 3: left upper antero-lateral area; 4: left lower antero-lateral area; 5: right upper posterior area; 6: right lower posterior area; 7: left upper posterior area; 8: left lower posterior area. Panel (**B**): representative LUS image showing B-lines. Panel (**C**): representative LUS image showing a lung consolidation.

**Figure 2 jcm-11-03032-f002:**
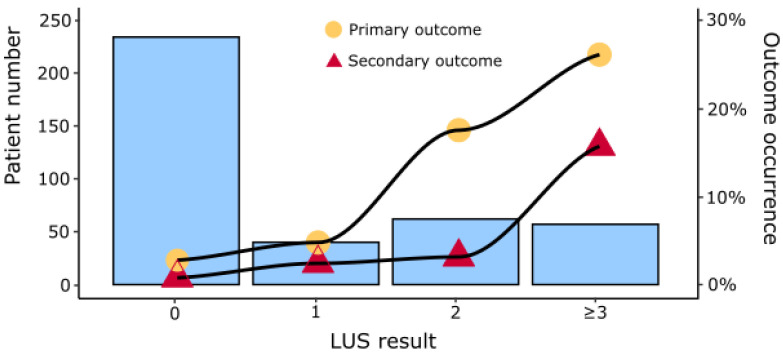
Patient number and primary and secondary outcome occurrence in patients stratified by LUS results.

**Figure 3 jcm-11-03032-f003:**
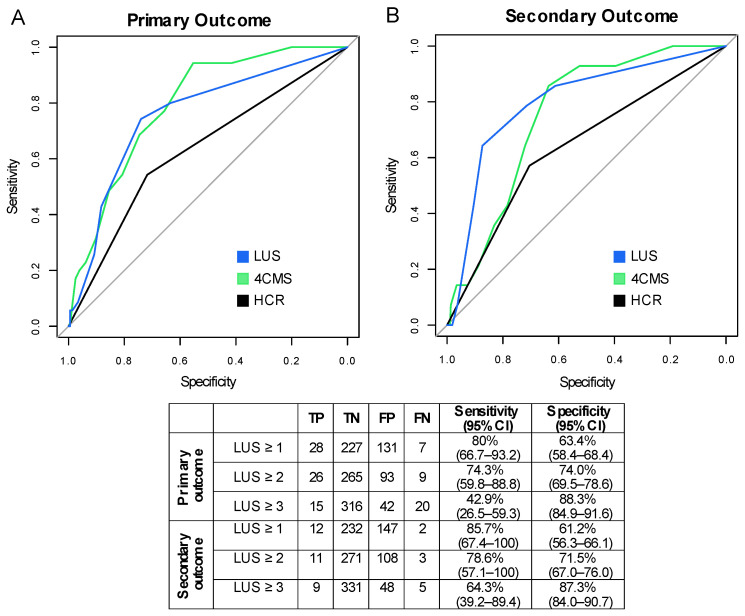
ROC curve analysis for prediction of (**A**) primary outcome and (**B**) secondary outcome. TP: True Positive; TN: True Negative; FP: False Positive; FN: False Negative.

**Figure 4 jcm-11-03032-f004:**
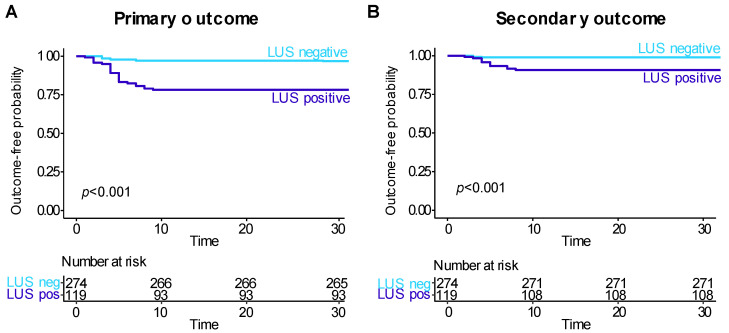
Kaplan–Meier curves for the (**A**) primary outcome and (**B**) secondary outcome in patients stratified by LUS results.

**Table 1 jcm-11-03032-t001:** Outcome prediction performance for the primary and secondary outcome.

		TP	TN	FP	FN	Sensitivity	*p*-Value *	Specificity	*p*-Value *
**Primary Outcome**	LUS positive	26	265	93	9	74.3% (59.8–88.8)	-	74% (69.5–78.6)	-
	HCR positive	19	257	101	16	54.3% (37.8–70.8)	0.17	71.8% (67.1–76.4)	0.52
	HCR positive and LUS positive	13	318	40	22	37.1% (21.1–53.2)	<0.001	88.8% (85.6–92.1)	<0.001
	HCR positive or LUS positive	32	204	154	3	91.4% (82.2–100)	0.03	57% (51.9–62.1)	<0.001
	High risk (4CMS ≥ 9)	8	336	22	27	22.9% (8.9–36.8)	<0.001	93.9% (91.4–96.3)	<0.001
	High risk (4CMS ≥ 9) or LUS positive	28	252	106	7	80% (66.7–93.3)	0.5	70.4% (65.7–75.1)	<0.001
**Secondary Outcome**	LUS positive	11	271	108	3	78.6% (57.1–100)	-	71.5% (67.0–76.0)	-
	HCR positive	8	267	112	6	57.1% (31.2–83.1)	0.45	70.4% (65.9–75)	0.78
	HCR positive and LUS positive	6	332	47	8	42.6% (16.9–68.8)	0.06	87.6% (84.3–90.9)	<0.001
	HCR positive or LUS positive	13	206	173	1	92.9% (79.4–100)	0.5	54.4% (49.3–59.4)	<0.001
	High risk (4CMS ≥ 9)	2	351	28	12	14.3% (0–32.6%)	0.004	92.6% (90.0–95.2)	<0.001
	High risk (4CMS ≥ 9) or LUS positive	11	256	123	3	78.9% (57.1–100)	1.0	67.5% (62.8–72.3)	<0.001

Legend. FN: False Negative; FP: False Positive; TN: True Negative; TP: True Positive. * calculated vs. LUS positive.

**Table 2 jcm-11-03032-t002:** Occurrence of primary and secondary outcomes in patient categories based on subjective evaluation alone (all patients), LUS, HOME-CoV (HCR), 4CMS or integration of LUS with HCR or 4CMS.

	N *	Primary Outcome	*p*-Value vs. All	*p*-Value vs. LUS	Secondary Outcome	*p*-Value vs. All	*p*-Value vs. LUS
All patients	393 (100%)	35 (8.9% [95%CI 6.5–12.1])	-	-	14 (3.6% [95%CI 2.1–5.9])	-	-
LUS negative	274 (69.7%)	9 (3.3% [95%CI 1.4–5.1])	<0.001	-	3 (1.1% [95%CI 0.4–3.2])	<0.001	-
HCR negative	273 (69.5%)	16 (5.9% [95%CI 3.6–9.3])	<0.001	0.003	6 (2.2% [95%CI 0.7–3.7])	0.01	0.04
HCR negative or HCR positive and LUS negative	340 (86.5%)	22 (6.5% [95%CI 4.3–9.6])	<0.001	< 0.001	8 (2.4% [95%CI 1.2–4.6])	<0.001	<0.001
HCR negative and LUS negative	207 (52.7%)	3 (1.4% [95%CI 0.5–4.2])	<0.001	<0.001	1 (0.5% [95%CI 0.02–2.7])	0.03	0.05
Low/intermediate risk (4CMS ≤ 8)	363 (92.4%)	27 (7.4% [95%CI 5.2–10.6])	<0.001	<0.001	12 (3.3% [95%CI 1.9–5.7])	0.33	<0.001
Low/intermediate risk (4CMS ≤ 8) and LUS negative	259 (65.9%)	7 (2.7% [95%CI 1.3–5.5])	<0.001	0.02	3 (1.2% [95%CI 0.4–3.3])	<0.001	0.68

* The % value in brackets corresponds to the rule-out efficiency, which can be calculated as (TN + FN)/(TP + FP + TN + FN). FN: False Negative; FP: False Positive; TN: True Negative; TP: True Positive.

## Data Availability

The data presented in this study are available upon reasonable request.

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
