# Peer review of "Lung Ultrasound Improves Outcome Prediction over Clinical Judgment in COVID-19 Patients Evaluated in the Emergency Department"

_jcm, 2022, doi:10.3390/jcm11113032_

Round 1

Reviewer 1 Report

Thank you for the opportunity to review your article “Lung ultrasound improves outcome prediction over clinical judgment in COVID-19 patients evaluated in the Emergency Department”

I consider this paper relevant as it evaluates a group of patients with COVID-19 patients with some relevant instrument such as an ultrasound point-of-care. 

However, in my opinion, some aspects have to be considered clarifying some aspects.

Methods:

  • I suggest identifying which are the 6 ED centers.
  • On page 4, about the LUS score:
  • Could you explain better how you scored each area? How many points did you assign for each finding?
  • You said that 12 areas were scanned, but posteriorly you said that 4 areas were evaluated. Could you explain better?

Discussion:

I missed some previous papers that showed a correlation between LUS and COVID-19 about the ability predicting outcomes, such as:

DOI: 10.1186/s13054-020-03416-1

DOI: 10.4187/respcare.08648

I suggest adding to the discussion.

Author Response

Point 1:

Methods:

  • I suggest identifying which are the 6 ED centers.

Response 1:

The characteristics of the participating centers are now detailed on page 3, lines 8-12.

 Point 2:

  • On page 4, about the LUS score:
  • Could you explain better how you scored each area? How many points did you assign for each finding?
  • You said that 12 areas were scanned, but posteriorly you said that 4 areas were evaluated. Could you explain better?

Response 2:

We would like to apologize for the insufficient clarity of the original version, and we thank the Reviewer for his/her comment. To improve method explanation and to allow reproducibility, we have now revised the entire section (page 4, lines 7-12), also adding a new explanatory figure 1. The modified LUS score, used to ease standardization in our study, was based on 8 and not on 12 areas (as in the original LUS score, reference number 19) for B-lines, with the addition of a “flat” score of 0, 3 or 6 points for consolidations (absent, unilateral or bilateral).

Point 3:

Discussion:

I missed some previous papers that showed a correlation between LUS and COVID-19 about the ability predicting outcomes, such as:

DOI: 10.1186/s13054-020-03416-1

DOI: 10.4187/respcare.08648

I suggest adding to the discussion.

Response 3:

The first reference was already cited in the Background section. In the same section, we have now added the second reference, as suggested.

Reviewer 2 Report

A large number of patients (521). The authors should describe in the result section the number of consolidations. "or any consolidation" in not clear. How do they see non-subpleural consolidations?

Author Response

A large number of patients (521). The authors should describe in the result section the number of consolidations. "or any consolidation" in not clear. How do they see non-subpleural consolidations?

Response:

We would like to apologize for the insufficient clarity of the original version, and we thank the Reviewer for his/her comment. For the ultrasound methods, we have now revised the entire section (page 4, lines 7-12), and added a new explanatory figure 1. The term “consolidation” has now been defined in the text, per study protocol. Since the term “non-subpleural consolidation” could be equivocal, we are now consistently using the term “consolidation” throughout the manuscript.

Only 10 patients had lung consolidations in our study, in line with a cohort of patients found clinically suitable for home discharge. This was added in the Results section (page 6 line 15).